# The amplifying influence of increased ocean stratification on a future year without a summer

J.T. Fasullo [1], R. Tomas[1], S. Stevenson[1], B. Otto-Bliesner[1], E. Brady[1] & E. Wahl[2]

In 1816, the coldest summer of the past two centuries was observed over northeastern North America and western Europe. This so-called Year Without a Summer (YWAS) has been widely attributed to the 1815 eruption of Indonesia's Mt. Tambora and was concurrent with agricultural failures and famines worldwide. To understand the potential impacts of a similar future eruption, a thorough physical understanding of the YWAS is crucial. Climate model simulations of both the 1815 Tambora eruption and a hypothetical analogous future eruption are examined, the latter occurring in 2085 assuming a business-as-usual climate scenario. Here, we show that the 1815 eruption drove strong responses in both the ocean and cryosphere that were fundamental to driving the YWAS. Through modulation of ocean stratification and near-surface winds, global warming contributes to an amplified surface climate response. Limitations in using major volcanic eruptions as a constraint on cloud feedbacks are also found.

[1] Climate and Global Dynamics Division, National Center for Atmospheric Research (NCAR), P.O. Box 3000, Boulder, CO 80307, USA. [2] Center for Weather and Climate (CWC), National Centers for Environmental Information, NCE325 Broadway, E/CC23, Boulder, CO 80305, USA. Correspondence and requests for materials should be addressed to J.T.F. (email: fasullo@ucar.edu)

On 10 April 1815, one of the largest volcanic eruptions of the last millennium occurred on the Indonesian island of Sumbawa[1,2]. This eruption of Mt. Tambora injected an estimated 60 Tg of sulfur dioxide into the stratosphere[3,4], where it converted to sulfate aerosol and spread globally. The associated increase in Earth's planetary albedo led to a sustained cooling of the surface for several years thereafter and is thought to have contributed to a broad range of follow-on effects across the globe[5–8]. Globally, up to 105,000 deaths are estimated to have occurred as a result of the eruption[6]. In its aftermath, cooling was pronounced in certain regions, including northeastern North America and central and western Europe[1,2,5,6], while in other regions, such as Southeast Asia, the response was the suppression of summer rainfall[9,10]. Due to its severe global-scale impacts and the availability of anecdotal accounts of the event, the eruption has been chosen as a test case for the ongoing Model Inter-comparison on the Climate Response to Volcanic Forcing (Vol-MIP) project[11].

While it is known that major regional climate anomalies followed the 1815 eruption, our understanding of the event and the potential implications of a similar future eruption can be improved through a more complete global-scale understanding of its climate response. Such an understanding is hampered by the limited availability of direct observations and the need to rely instead on proxy-based reconstructions that are relatively uncertain in relation to instrumental data. Moreover, some early studies using coupled climate models relied on a small number of ensemble members[12,13], making it a challenge to separate the forced climate response from internal variability, an issue that has been addressed more thoroughly for other eruptions in recent work[13,14]. In addition, simulated responses to major eruptions are themselves suspected of being biased in key respects[15], although forcing uncertainties and the transient evolution of the climate response make these biases difficult to estimate[4,9,16–20]. Addressing these issues is critical to understanding the large-scale processes that drove the 1815 climate response and is a key part of assessing the potential impacts of a future eruption.

Here, we assess a newly available ensemble of climate simulations spanning the last millennium, the Community Earth System Model (CESM) Last Millennium Ensemble (LME[21]), which bypasses some of the above concerns due to its multiple members and use of a single climate model. These simulations use the Coupled Model Intercomparison Project (CMIP5) climate forcing reconstructions[22]. Volcanic forcing is specified from ice-core-derived estimates of aerosol loadings as a function of latitude, altitude, and month[23] while stratospheric aerosols are prescribed as a fixed single-size distribution in the three layers in the lower stratosphere. The Tambora eruption coincided with a minimum in solar activity (the so-called Dalton minimum) and changes in total solar irradiance are prescribed, with an estimated 11-year solar cycle imposed[24].

Our results show that the 1815 eruption of Mt. Tambora triggered strong responses at the land surface and within the atmosphere and ocean. The YWAS can thus best be understood

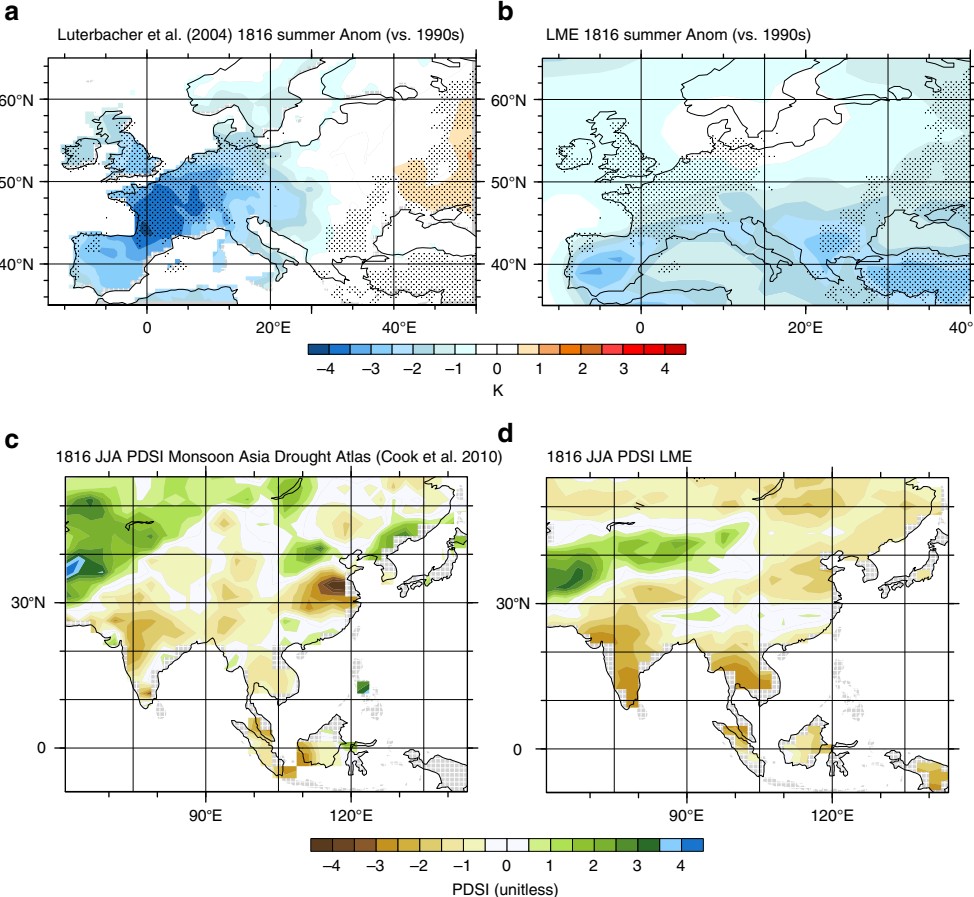

**Fig. 1** Proxy reconstructions and LME-simulated responses compared. European 1816 summer temperature anomalies relative to 1990–1999 in **a** proxy reconstructions[26] and **b** from the LME, and **c** southeast Asian Palmer Drought Severity Index (PDSI), an indicator of drought[10], and **d** their ensemble means over contemporaneous intervals from the LME. Stippling indicates regions where reconstructions lie outside of the ensemble model range for both temperature and PDSI

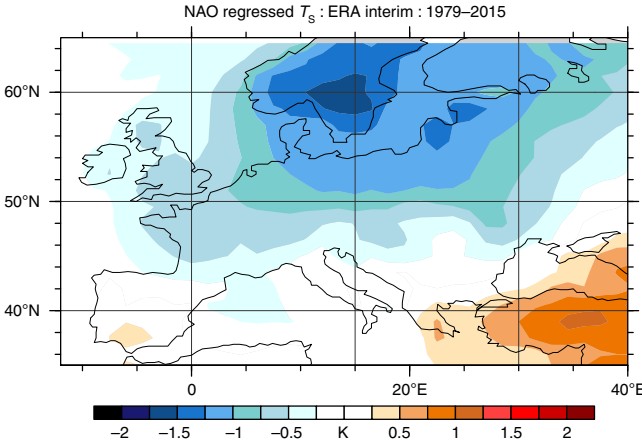

NAO regressed $T_S$ : ERA interim : 1979–2015

**Fig. 2** Regional temperature structure of the NAO. Regression of the negative phase of the NAO during June through August against contemporaneous surface temperature for the region considered in Fig. 1 (see "Methods"). Contributions to cooling from the negative phase of the NAO are greatest in regions where the model forced response fails to reproduce the cooling estimated in reconstructions

as resulting from the combined influences of the direct volcanic aerosol-driven radiative cooling of the surface modulated by these responses. In the cryosphere, key responses include the amplifying effects of increased snow cover and sea ice extent, while in the ocean they include the dampening of surface cooling due to substantial ocean heat content (OHC) release. A warmer climate is found to modulate all of these responses with the net effect of amplifying the surface cooling response, due largely to an increase of upper ocean stability and a decrease of its overall moderating influence.

## Results

**Evaluation of the 1816 Year Without a Summer in the Last Millennium Ensemble**. Model performance is evaluated (see "Methods" section) with satellite retrievals and reanalyses based on late 19th and 20th century volcanic eruptions. The LME is found to reproduce key features of the satellite-estimated net top-of-atmosphere (TOA) radiative imbalance arising from the recent 1991 eruption of Mt. Pinatubo[25] and basic aspects of volcanic responses in June through August (JJA) in the year following eruptions estimated from reanalyses, Here, we also use updated reconstructions in Europe and Asia (Fig. 1, refs. [10,27]), where anecdotal accounts suggest the eruption respectively drove strong cooling and drought, to evaluate model fidelity.

**Simulating the 1816 Year Without a Summer**. Reconstructions spanning JJA of 1816, the YWAS, show regions of agreement and disagreement with the LME forced response, obtained by averaging across 18 volcanically forced LME ensemble members (13 fully forced and 5 volcanic-only forced, Fig. 1). Reconstructed surface temperature anomalies are characterized by cooling as large as −4 K in western Europe that exhibits strong zonal structure and becomes small east of about 20°E, with warming in eastern Europe. The LME forced response exhibits both similarities, such as cooling in western Europe, and differences, including only modest cooling in northern Europe and a general lack of warming in eastern Europe. These contrasts between reconstructions and the forced response may arise from error in the reconstructions, error in the model forcing[18] and forced response, or the influence of internal variability[16,18]. Given the uncertainties arising from internal variability and errors in both reconstructions and models neither can be taken as definitive[28]

and so the goal here is to assess the general consistency between the two. This reconstruction is based on a data set that includes a large number of homogenized and quality-checked instrumental data series, a number of reconstructed sea-ice and temperature indices derived from documentary records for earlier centuries, and a few seasonally resolved proxy temperature reconstructions from Greenland ice cores and tree rings from Scandinavia and Siberia[27]. A separate spatial reconstruction of European 1816 summer temperatures, based solely on instrumental data, indicates anomaly patterns much like the reconstruction shown in Fig. 1a[26], suggesting that model forcing and error, and the influence of internal variability, are the most salient sources of contrast, whereas other reconstructions[29] agree more closely with the model forced response, suggesting a role for internal variability. Regarding the latter, a role for the negative phase of the North Atlantic Oscillation (NAO)[30] and its observed summer teleconnections (Fig. 2) is suggested as it contributes to anomalous cooling in northern Europe and enhances the zonal temperature gradient, helping reconcile these model-reconstruction differences to a degree. The LME mean NAO anomaly in the summer of 1816 (i.e., its forced response) is small but statistically significant (−0.21 ± 0.18, 5% conf.), while the actual value in nature varies considerably across reconstructions, particularly in summer. Uncertainties surrounding the models, proxies, and NAO reconstructions preclude a precise determination of its contribution.

In southeast Asia, hydroclimate reconstructions[10] show regions of strong drought spanning India, Indonesia, and southeast Asia, bounded to the north by wetter than normal conditions in Pakistan and western China. While uncertainty surrounding these reconstructions also remains considerable[28,31], the degree of agreement with the LME forced response is notable particularly given the large spread across individual realizations in the model (see Supplementary Discussion and Supplementary Figs. 1, 2).

Together the reconstructions in Fig. 1 suggest the skillful reproduction of key aspects of the YWAS climate response in the LME, particularly for Asian drought and cooling in portions of southern Europe, and underscore the eruption's global reach and the need to develop a suitable large-scale process-based understanding of the event in order to begin to understand its modulation in a warmer climate. The LME simulations provide considerable insight into the main energetic responses governing surface impacts. For example, a useful measure of the eruption's radiative effect is provided by the global clear-sky shortwave flux through the atmosphere (SW-C$^{atm}$), computed as the difference between clear-sky downwelling solar radiation at the surface and TOA. This is shown in conjunction with simulated global mean 2-m air temperature ($T$) and precipitation ($P$) anomalies (Fig. 3a), and associated TOA radiative flux anomalies (Fig. 3b). In the months following the eruption, a decrease in SW-C$^{atm}$ of about 16 W m$^{-2}$ is simulated, initially in the tropics but spreading globally in the months thereafter (Supplementary Fig. 3). The decrease in TOA absorbed shortwave radiation (SW) exceeds that in outgoing longwave radiation (also Supplementary Fig. 4) by a factor of about 2, such that a net TOA deficit of about 6.5 W m$^{-2}$ is realized in mid-1815. These simulated radiative responses are in general agreement with those derived in other recent model ensembles[32,33]. In response to this radiative deficit, the surface cools and the hydrologic cycle weakens. By late 1815, $T$ drops by 0.74 K in the ensemble mean, an amount on par with the climate system's net warming in the entire 20th century, and rainfall drops by 0.085 mm day$^{-1}$ (2.8%), as less sunlight reaches the surface and reduces the energy available for evaporation.

Cooling is particularly strong in the Northern Hemisphere high latitudes (Fig. 4a, b), where cryospheric responses, including increases in snow cover over land and sea ice over ocean, extend

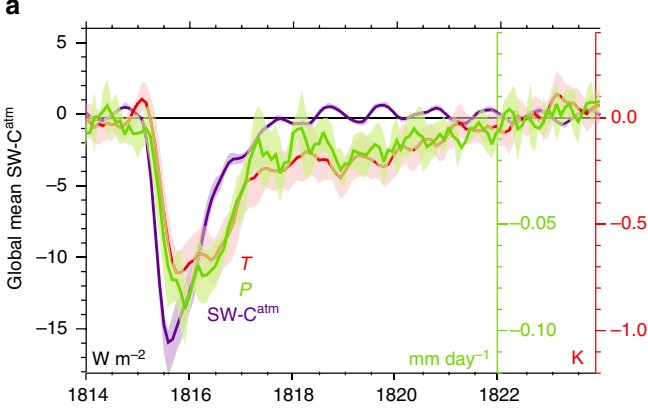

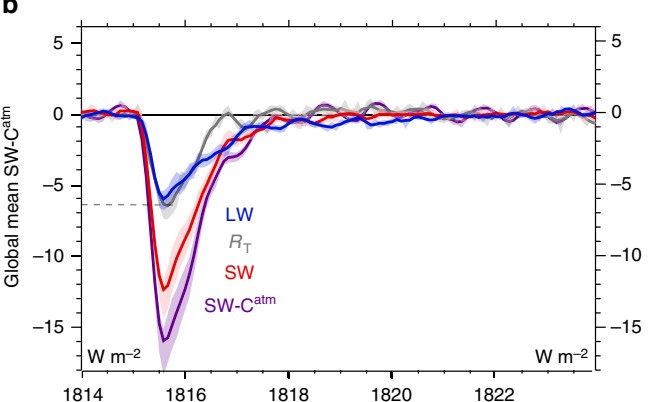

**Fig. 3** Global mean-simulated responses to the 1815 eruption. Simulated anomalies relative to 1814 are shown in **a** global mean clear-sky shortwave flux through the atmosphere (SW-C$^{atm}$), 2-m air temperature ($T$), and precipitation ($P$), and **b** SW-C$^{atm}$, global mean TOA incoming net ($R_T$), shortwave (SW), and outgoing longwave (LW) radiation. Shading denotes the range of two standard errors. Years indicated on the abscissas begin at the corresponding tick marks

the persistence of aerosol-induced albedo increases (Supplementary Fig. 5). The simulated anomalies are broadly consistent with other recent model ensembles in their depiction of pervasive northern hemisphere cooling over land[32,33]. Anecdotal accounts of the YWAS associate cool temperatures with the lingering radiative effects of volcanic aerosols[6] but in the LME, SW-C$^{atm}$ anomalies have largely dissipated by the summer of 1816 (Fig. 3, Supplementary Fig. 3). Rather, the simulations show that it is mainly a cold upper ocean, associated land amplification (discussed further below), and other responses that set the stage for the anomalously cool YWAS conditions. In addition to cryospheric changes, a forced El Niño event (Fig. 4b) is realized (in 12 of the 18 ensemble members) and it is associated with positive rainfall anomalies over land (Fig. 4c), contributing to additional cooling in the western US and southern Europe.

Despite the severity of the simulated weakening in the hydrologic cycle, energy budget perturbations suggest that impacts of the eruption may have been much larger had it not been for the mediating role of the ocean. For example, a peak net TOA flux anomaly of −6.5 W m$^{-2}$ is simulated following the eruption (Fig. 3b), a deficit that is roughly equal to 8% of the global mean evaporation (~85 W m$^{-2}$) that drives the water cycle, yet the cycle's weakening is simulated to be less than 3%, an amount on par with the simulated weakening in another recent ensemble[33]. One potential explanation for the discrepancy is that responses in other terms of the surface energy balance, such as in

sensible heat or longwave radiation, compensate for the imposed surface shortwave perturbation and mitigate the surface energetic imbalance so as to lessen reductions in evaporation. However, in these other terms, simulated responses (shown in Fig. 5) act to accentuate shortwave cooling anomalies rather than lessen them, though the magnitude of each is relatively small.

LME simulations demonstrate instead that, as suggested by land-sea contrasts in the temperature response (Fig. 4a), ocean cooling plays a key role in mediating the eruption's surface response, with a peak OHC reduction of $7 \times 10^{22}$ J in 1817 (Fig. 5), followed by a gradual recovery, particularly in the deep ocean, lasting over a decade (Figs. 5, 6, Supplementary Fig. 6). The associated heat flux out of the ocean in the years following the eruption thus played a key role in sustaining global evaporation near pre-eruption levels and limiting the reduction in global mean rainfall. The long timescale of OHC recovery suggests the potential for a continuing influence of the earlier 1809 eruption[34] on OHC contemporaneous with the 1815 Tambora response. LME simulations show that the associated surface anomalies had largely dissipated by 1815, however.

The simulated ocean cooling response is particularly pronounced at the surface in the subtropics (Fig. 6, Supplementary Fig. 6) and extends to depth on the poleward fringes of the subtropical overturning cells, where subduction and ventilation in the upper 1000 m are strong[35] (Supplementary Fig. 7). In the upper 100 m, cold anomalies peak in early 1816 (Fig. 6, Supplementary Fig. 6) and cooling extends to depth (Fig. 6), underpinning the cool surface, particularly equatorward of the main YWAS regions in North America and Europe (Fig. 4b, Supplementary Fig. 6). Surface ocean temperatures drop disproportionately in the Northern Hemisphere (Fig. 4a, b), where integrated OHC anomalies are comparable to those in the Southern Hemisphere but ocean extent is less.

The forced response of LME simulation to the 1815 eruption reproduces many of the YWAS accounts, such as drought in Asia and cooling in northeastern North America and western Europe (Fig. 4), and suggests several others. By January through March (JFM) of 1816, land regions south of 60°N are simulated to have cooled generally, with particularly strong cooling in central North America and southern Eurasia. The Aleutian low had also strengthened, and the tropical Walker circulation had weakened (Supplementary Figs. 8, 9). The weakened Walker circulation is consistent with strong reductions in rainfall surrounding the Maritime continent (Fig. 4, Supplementary Fig. 9) and a transition in the atmosphere, such that during the YWAS, El Niño SST anomalies, and associated cloud anomalies emerge[16] (Supplementary Figs. 8, 10). On average, the response of clouds to the eruption is characterized both by regional anomalies, associated for example with a transition into El Niño conditions in 1816–1817 (Supplementary Fig. 10), and a global reduction in cloud amount (Table 1) and, specifically tropical high cloud amount, implying a positive feedback in response to the eruption due to increased LW emission to space.

**Simulating an analogous 2085 eruption**. The large-scale perturbations driven by Tambora's 1815 eruption (as demonstrated above for example in its oceanic, surface, and atmospheric responses) raises the question as to what the modulating influence of climate change might be if such an event were to unfold in the future. Here, we therefore compare and contrast key large-scale aspects of a hypothetical 2085 eruption (see "Methods") with the 1815 response (Fig. 7). Many aspects of the radiative response closely resemble those during the 1815 event and these include a dramatic decrease in SW-C$^{atm}$, an increase in clear-sky albedo at low latitudes (Supplementary Fig. 11), and reductions in

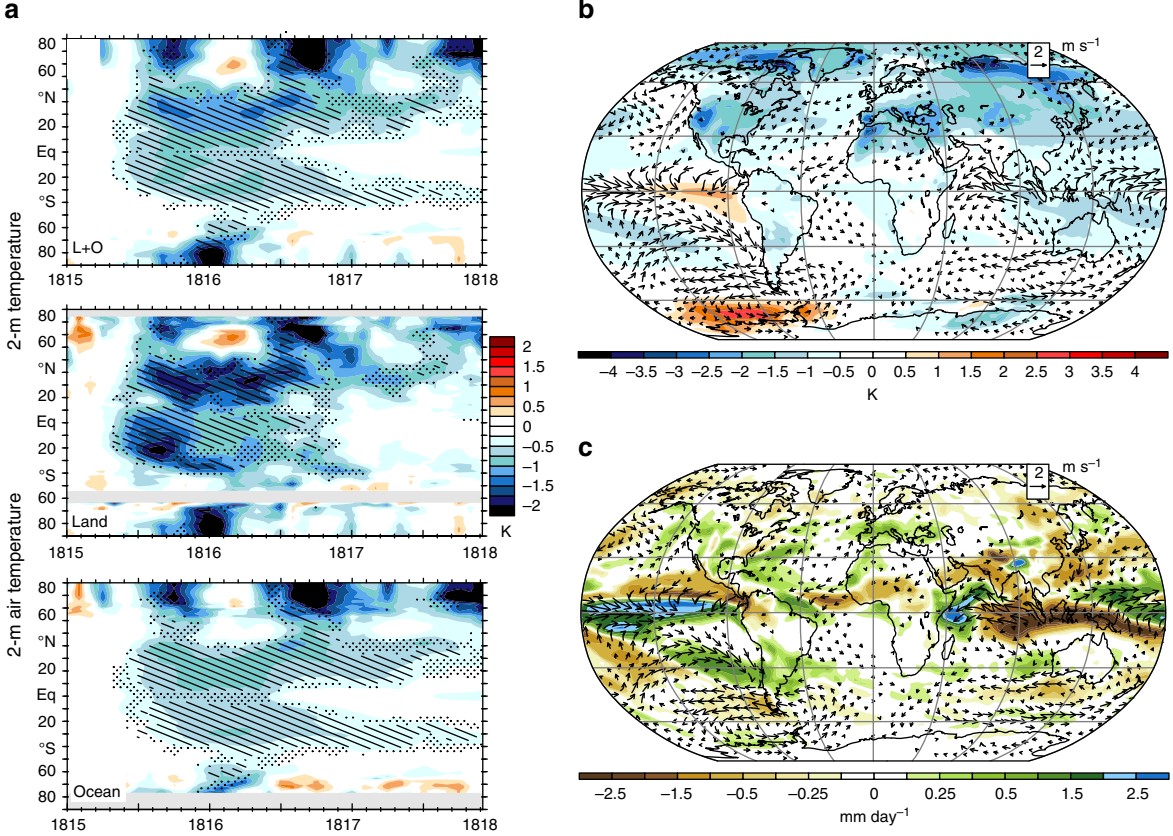

**Fig. 4** Regional-simulated responses to the 1815 eruption. **a** Zonal mean *T* for land and ocean (L + O), and land, and ocean, separately and YWAS July through September near-surface winds and **b** *T* and **c** *P*. Mapped colors (**b**, **c**) and hatching (**a**) indicate anomalies exceeding twice the ensemble standard error. Stippled regions in **a** indicate anomalies between one and two times the standard error. Years indicated on the abscissas begin at the corresponding tick marks

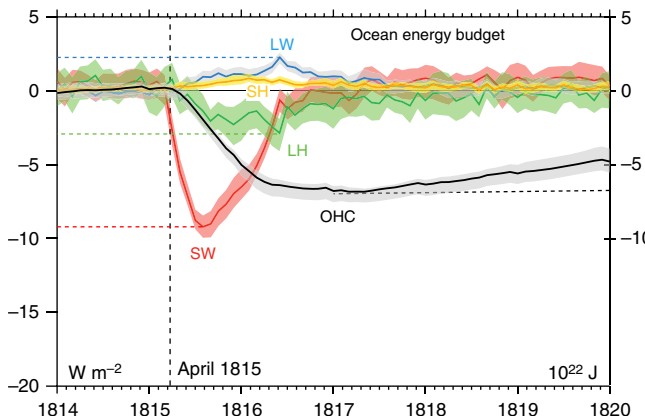

**Fig. 5** Responses in terms of the surface energy budget and OHC to the 1815 eruption. Changes in terms of the surface energy budget over ocean (left axis) following the 1815 Mt. Tambora eruption including net upwelling longwave flux (LW), sensible heat (SH), latent heat (LH), and net surface shortwave absorption (SW). Also shown is the change in full-depth OHC (right axis)

TOA net SW and LW (Supplementary Figs. 12, 13) fluxes. Differences in the overall magnitude and spatial structure of the SW radiative responses are generally small. At the surface, however, a future eruption's peak response in both global mean temperature and rainfall is simulated to be approximately 40% greater than in 1815 (Fig. 7a), though the uncertainty associated with the ensemble spread is also considerable and limits a precise

quantitative estimation of the increase. Changes in the background climate state are likely to be instrumental in driving this increase. Key among the background changes is the fact that the upper ocean has warmed considerably in the simulated future climate, particularly in the tropics, contributing to an increase in ocean stratification (Fig. 7b). Much of the increase in surface and upper ocean cooling following the 2085 eruption is simulated at low latitudes in winter and spring, and in northern polar regions, where the cryospheric response increases in magnitude and is displaced poleward due to the warmer base state, thereby moderating its radiative impact such that the net change in global surface albedo is small (Supplementary Fig. 14). As a consequence of increased ocean stratification, temperature anomalies in the upper ocean do not penetrate to depth as efficiently as in 1815 (Fig. 7d), leading to greater surface cooling and reduced cooling at depth at all latitudes except from 30° to 45° in both hemispheres. Increases in upper-ocean stability are evident on a global scale and persist throughout the annual cycle. Changes in wind speed, salinity, and mixed layer depth are also likely to influence the depth of cooling, though the spatial distribution of these fields is more complex and seasonally dependent than for stability (Supplementary Figs. 15, 16). Quantifying the relative roles of each in modulating the volcanic response remains a work in progress. At high latitudes, SW responses strengthen during summer as the polar oceans have moved toward a more ice-free state allowing the formation of sea ice induced by the eruption to have a disproportionate effect on albedo (Supplementary Fig. 11). Despite its enhanced magnitude relative to the 1815 eruption, the surface cooling induced by a 2085 eruption (~1.1 K, Fig. 4a) remains much less than the associated warming of the

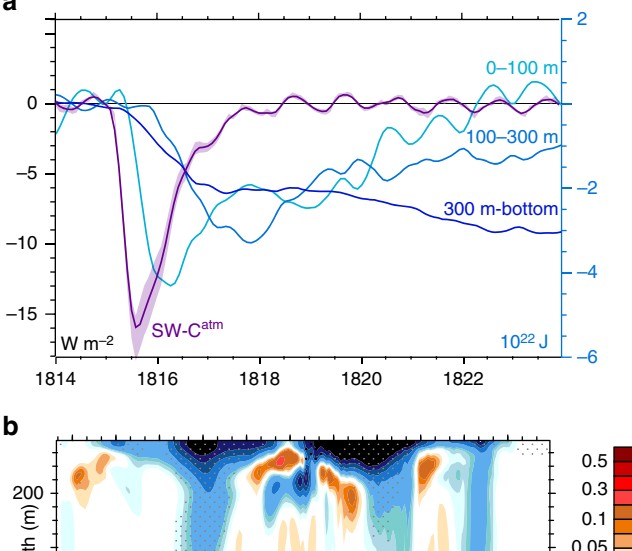

**Fig. 6** Responses in OHC with depth to the 1815 eruption. **a** Evolution of SW-C$^{atm}$ (left axis) and OHC (right axis) across various ocean layers. **b** Anomalies in zonal mean ocean temperature by depth and latitude in 1816, with stippled regions indicating a consistent sign of anomalies across at least 14 of the 18 (80%) ensemble members. Shading in **a** denotes the range of two standard errors. Years indicated on the abscissas begin at the corresponding tick marks

**Table 1 Cloud feedbacks under climate change and in response to the Tambora eruption**

| Cloud changes | Tropical (30°–30°) CC: YWAS | Mid-latitude (30°–60°) CC: YWAS | High latitude (60°–90°) CC: YWAS |
|---|---|---|---|
| Total cloud | −0.41: 1.26 | 0.77: 2.76 | 0.06: −0.26 |
| High cloud | −0.14: 2.33 | 0.24: 4.61 | 0.43: 1.16 |
| Middle cloud | −0.69: −0.24 | 1.21: 2.13 | −0.16: −1.14 |
| Low cloud | −0.40: 0.08 | 0.83: 0.62 | 0.17: −0.82 |

*Note*: Changes in annual−mean cloud percent per degree of global surface temperature change (for anthropogenic climate change (CC), left, and YWAS anomalies, right) by cloud type and latitude bands integrated over both hemispheres. For anthropogenic climate change, the ensemble mean difference of fields between 1815 and 2085 is used, while for the Tambora eruption anomalies in 1816 relative to the full 19th century are used. See also Supplementary Fig. 10

eruptions and transient background climate change, large contrasts are identified. Generally, simulated cloud changes are much greater per degree cooling following volcanic eruptions than those that accompany anthropogenic climate change. For example, under greenhouse warming, estimated from 1815 to 2085 differences, total tropical cloud amount decreases by −0.41% K$^{-1}$ yet following volcanic eruptions, it also decreases (with cooling) by about 1.26% K$^{-1}$ and changes are thus not only substantially different in magnitude but opposite in sign. For tropical high cloud the contrast is particularly large with reductions with greenhouse gas warming of about −0.14% K$^{-1}$, whereas following volcanic eruptions high clouds decrease in extent by 2.33% K$^{-1}$. As the primary uncertainty in determining climate sensitivity lies in quantifying the response of such clouds to warming[38,39], these discrepancies suggest that attempts to constrain cloud feedback and climate sensitivity uncertainties through observations of cloud and radiative responses following volcanic eruptions is unlikely to provide a useful constraint. Volcanic eruptions may, however, offer insight into the capabilities and limitations of models in evaluating geoengineering strategies and associated follow-on effects. Continuing to confront climate models with a range of climate phenomena across timescales, including major volcanic eruptions of the last millennium, remains a useful exercise for improving our understanding of climate as it changes beyond the range of observed variability in Earth's recent history.

## Discussion

The evolution of YWAS climate anomalies is explored here in the context of simulated large-scale changes arising from the climate system's coupled response to the exceptionally strong transient radiative forcing of Mt. Tambora's 1815 eruption. While these changes and their modulation by a warming background state are evident in all components of the climate system, the damping of the response by the upper ocean is identified as key, both in understanding the timing and magnitude of near-surface anomalies during the YWAS and the amplification of the large-scale response under transient warming. Given the robust depiction of these processes across current coupled models[40,41], it is likely for this amplification to be depicted consistently across models.

Questions nonetheless remain across a range of issues. For example, the relatively small number of ensemble members available for the 2085 eruption limits precise statements regarding the magnitude of response amplification due to changes in the background state (Fig. 7). Details of the regional responses to eruptions also remain uncertain and these may exhibit model dependence, including differences in the response of internal modes (e.g., such as the El Niño—Southern Oscillation, or ENSO, and the NAO), cryospheric feedbacks, and clouds. Biases in the simulated mean state and responses, particularly in clouds and stratospheric dynamics, remain a major concern regarding model fidelity, even in the CESM, and efforts to remediate these remain ongoing. Questions surrounding the magnitude and timing of stratospheric aerosol loading[41,42] are also a continuing source of uncertainty as different loading estimation methods can contrast considerably with each other and continue to evolve and improve over time. The influence of climate change on stratospheric aerosol injection and loading also continues to be understood and is not accounted for here[43].

Increased confidence in our understanding of both the 1815 event and the potential response to analogous future eruptions is a likely outcome of ongoing efforts to address the issues above. The generation of large ensembles imposing targeted forcings for specific events will aid in identifying statistically significant forced

background state (4.2 K) and hence the magnitude of this amplified, historically exceptional volcanic event is nonetheless unable to offset even a third of the accumulated transient background warming since the 1815 Tambora eruption.

Lastly, it is worth noting that volcanic eruptions have been proposed previously as a means of understanding the climate system's base sensitivities. One key context relates to climate sensitivity and cloud feedbacks in response to forcing, such as for example increases in greenhouse gases[36,37]. Here, responses in the cloud field to volcanic eruptions and a warming climate are found to differ significantly in key respects. By regressing cloud fields against global mean surface temperature (Table 1) during both

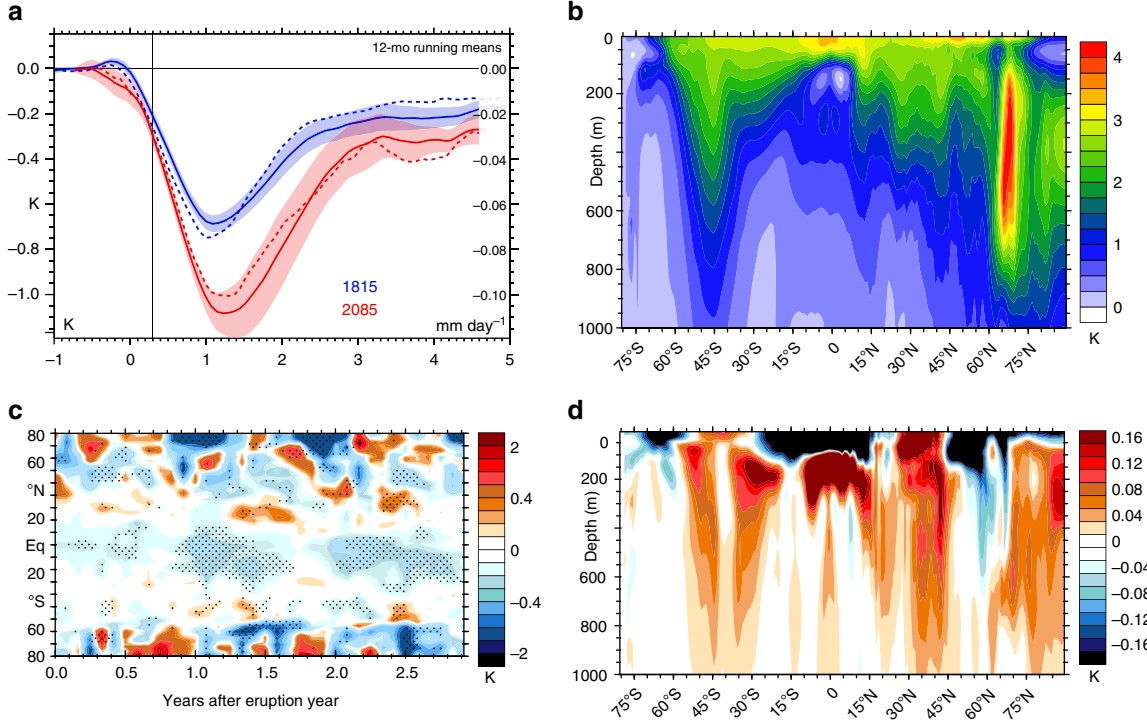

**Fig. 7** Changes in eruption responses in surface and ocean temperatures between 1815 and 2085. Ensemble mean-simulated **a** anomalies in global mean surface temperature (K) and rainfall (mm day$^{-1}$, dashed, right axis) for 1815 (blue) and 2085 (red) eruptions, smoothed with a 12-mo running mean and shown with a 2 standard error range for temperature (shading), **b** annual mean net ocean warming by latitude and depth estimated from the 2085–2094 change from 19th century ensemble zonal mean, **c** differences in ensemble zonal mean 2-m air temperature anomalies between the 2085 and 1815 eruptions (stippled where anomalies exceed twice the standard error), and **d** the 2086–1816 difference in ensemble zonal mean ocean temperature anomalies by latitude and depth. Years indicated on the abscissas begin at the corresponding tick marks

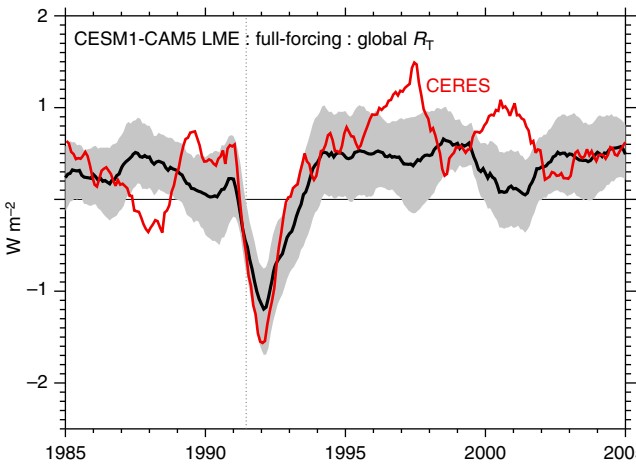

**Fig. 8** Validation of the LME-simulated TOA flux response to the 1991 Pinatubo eruption. LME mean global TOA net flux ($R_T$) vs. CERES (red) from 1985 to 2005 and spanning the 1991 eruption of Mt. Pinatubo. A 12-month running mean is applied to simulated and observed fluxes prior to plotting and computing ensemble spread. Gray shading indicates the standard deviation range in the LME

changes on regional scales. Establishing confidence in these ensembles will depend also on advances in model skill, given the presently large inter-model spread that exists. Further exploration of the interactions between forcing and internal variability in historical and transient warming climate states with a fully resolved stratosphere is also likely to provide key insight. An adequate estimation of socioeconomic impacts arising from future eruptions, such for example on agriculture[39], depends

critically on addressing these outstanding issues related to regional-scale prediction. As a complement to model development activities, improved proxy reconstructions and reanalyses are also likely to play a key role in narrowing these uncertainties.

Irrespective of these uncertainties in models and forcing data sets, a number of additional questions remain. For example, a broad range of potential future eruption characteristics exists that has yet to be thoroughly explored, including the latitude, temporal evolution, season, and aerosol characteristics of such an eruption. Some aspects of climate change's influence on the response are likely to be robust to such aspects, such as the enhancement of the surface temperature response due to increased upper ocean stratification. However, the influence of other aspects, such as the seasonally and spatially dependent background wind field and mixed layer changes, and interactions with internal modes of variability are less obvious[16,17,44,45]. Clarifying these aspects remains a key science goal.

## Methods
**Model runs**. The analysis of the 1815 eruption of Mt. Tambora is based on the CESM1-CAM5-LME[21], which now incorporates 13 fully forced simulations differing only in their initial conditions at 850 CE. The model resolution is nominally 2° for the atmosphere and land components (144 × 96) and 1° for the ocean (320 × 384). In the ensemble, 18 members include volcanic forcing (as there are an additional five volcanic-only forced members) and these are therefore the focus of this analysis. Volcanic aerosol effects in the model are confined to radiative impacts and are otherwise non-interactive. Specifically, the aerosols do not interact with tropospheric clouds[38]. Twelve additional ensemble members extending from 2005 to 2100 have been generated for estimating future Tambora effects using forcings specified under Representative Concentration Pathway 8.5 (RCP8.5)[21], four of which also include an analogous 2085 eruption not specified in the default RCP8.5. While there are fewer members included in these simulations, the explicit availability of simulations with and without volcanic aerosols aids in the computation of the forced response to volcanic forcing, as members with and without such forcing can be differenced to account for the large transient changes that accompany the

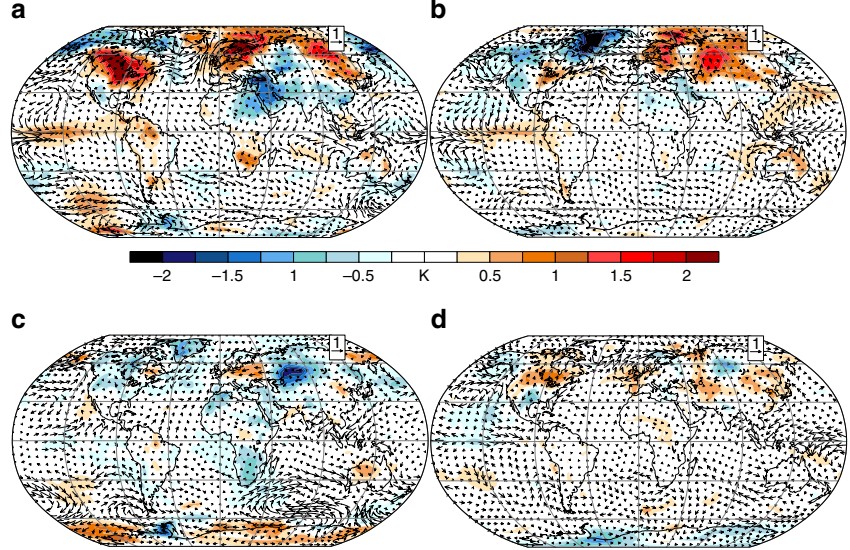

**Fig. 9** Composite regional responses to strong and weak eruptions of the late 19th and 20th centuries. Anomalies in 2-m air temperature and 10-m winds from NOAA's 20th century reanalysis for eruptions after 1880 (Table 2) for **a** JFM and **c** JJA in the year after the four strongest eruptions (**a, c**) and five weaker eruptions (**b**-JFM, **d**-JJA). Anomalies are computed by removing decadal and longer time series using the decadal filter described in ref. [47], a step necessitated by the transient background state, and temperature anomalies are plotted only where they exceed 95% significance (assessed using a Monte-Carlo technique). The dissimilarities in the fields, particularly in the Arctic, suggest it may be inappropriate to include weaker eruptions in a composite for the purposes of model validation, as the variability they contain is unlikely to be strongly forced

### Table 2 Volcanic eruptions of the late 19th and 20th centuries

| Eruption date | Latitude | Stratospheric SO$_2$ mass (Tg)[a] |
|---|---|---|
| **27 August 1883** | 6°S | 44 |
| *10 June 1886* | *38°S* | *4–5* |
| *15 July 1888* | *38°N* | *3–4* |
| **24 October 1902** | 15°N | 30 |
| *10 April 1932* | *36°S* | *3* |
| **17 March 1963** | 8°S | 20 |
| *10 October 1974* | *14°N* | *4* |
| *4 April 1982* | *17°N* | *7* |
| **15 June 1991** | 15°N | 20 |

[a] From stratospheric SO$_2$ injection data from refs. [49–52]
Volcanic eruptions considered in ref. [48]. Eruptions classified as being "strong" and "weak" are shown in bold and italicized text, respectively

RCP8.5 background state. Potential changes in the stratospheric injection mass and associated lifetime of aerosols arising from climate change may be important[43] but are not considered in these ensembles. Rather the focus here is on the response of the coupled climate system to forcing comparable to the 1815 eruption and associated changes arising from the warming base state.

A key issue in assessing the plausibility of the simulated response to volcanic eruptions in the CESM is the validation of the TOA radiative response. To assess this response, the net flux ($R_T$) from 1985 to 2005 for the nominally 1° version of the CESM was evaluated with reconstructed CERES $R_T$ estimates[46]. A very strong correspondence was found. Here, the ensemble mean evolution of $R_T$ using the 2° CESM-LME as described in ref. [21] from 1985 to 2005 is compared with CERES (Fig. 8), where a 1 standard deviation ($\sigma$) range among ensemble members is also shown (shading). In general, agreement between the ensemble and observations is strong, with CERES values lying beyond the 1-$\sigma$ range mainly during major ENSO events, such as during 1987–1988, 1997–1998, and 1999–2001. The magnitude of the ensemble mean perturbation in $R_T$ contemporaneous with the eruption is slightly weaker than in observations, though CERES estimates are well within the ensemble range (Fig. 9).

Dynamic responses of CMIP5 models to eruptions of the late 19th and 20th centuries are examined in ref. [15] (hereafter D12), where they are identified as being systematically biased in the winters following eruptions, relative to 20th century reanalysis fields[48]. The biases are claimed to relate mainly to an observed high latitude warming response in the Arctic, which D12 associate with the NAO

(Fig. 2) and find to be absent from most models. In their analysis, D12 include nine eruptions in total (Table 2), of which five are small eruptions as measured by their stratospheric SO$_2$ injection (<8 Tg). An open question is whether the inclusion of the small eruptions in their composite is warranted, due to the potentially minor climate effects of these eruptions, and whether their conclusions regarding model biases are therefore justified. Here, we conclude that the inclusion of the small eruptions is not warranted as major contrasts are found between composites of the anomalies following the four strong (bold in Table 2) and five weak (italics) eruptions, including the temperature response over North America, Greenland, and central Eurasia. In addition, the strongest eruptions all occurred within 15° of the equator, where Mt. Tambora resides, whereas several of the weaker eruptions occurred outside of the tropics, thereby raising the issue of the response's dependence on latitude[53–55]. The need to address volcanic magnitude, latitude, and timing in validating the model response highlighted here is also in agreement with the recent work of others[14,20]. Given these limitations, the extent to which climate models systematically fail to resolve key volcanic responses remains unclear.

The NAO reconstruction used for Fig. 2 is from ref. [56] and 2 m temperature from ECMWF's ERA Interim reanalysis[57].

**Anomaly calculations**. The methods used to compute anomalies for 19th and 21st century eruptions differ due to the contrasting nature of available simulations and transient background changes. For the 1815 eruption, unless specified otherwise anomalies are estimated relative to the 19th century, a period in which transient climate changes remain small relative to the 20th century. In instances where the transient global mean changes of the response are a focus, rather than its spatial structures (e.g., Fig. 2a), anomalies are computed relative to 1814, due to the lingering recovery from the 1809 eruption that would otherwise be conflated with that of Mt. Tambora[32]. Of note are recent simulations that quantify contributions of the 1809 eruption and find the effect that persists through 1816 to be that of a cooling that is relatively small and confined mainly to high latitude land regions[32]. In contrast to the anomaly computations for 1815, and as multiple simulations for the 21st century are available that explicitly include or omit the aerosol forcing of a Tambora-like eruption, the eruption's forced response can be quantified directly from the difference of the two sets of simulations and this approach is therefore used. This approach is also necessitated by the large concurrent transient background climate changes under RCP8.5 that accompany the hypothetical future eruption.

**Data availability**. The data that support the findings of this study are publicly available. The CESM-LME outputs are available via the Earth System Grid (www.earthsystemgrid.org) as single variable time series in self-documenting loss-less compressed Network Common Data Form (NetCDF-4) format. Proxy reconstruction data are available at the National Centers of Environmental Information; https://www.ncdc.noaa.gov/data-access/paleoclimatology-data/datasets/climate-

reconstruction. Simulations of future Tambora eruptions are available from the corresponding author upon reasonable request.

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

## Acknowledgements

Author affiliations include the Dr. Fasullo's participation in this work, which was supported through NSF Award ID AGS 1243107, NASA Award Number NNH11ZDA001N, and DOE Award ID DE-SC0012711. The authors would also like to acknowledge the contributions of three reviewers.

## Author contributions

Dr. Fasullo envisioned and conducted the analysis and led the writing and revision of the manuscript. Dr. Otto-Bliesner designed and supervised production of simulations of the CESM-LME, while Dr. Tomas designed and performed the simulations of future Tambora eruptions. Drs. Stevenson, Brady, and Wahl assisted in manuscript revisions and applying the paleoclimate record to the CESM simulations.

## Additional information

**Competing interests:** The authors declare no competing financial interests.

