## [Peer Review File · Nature Communications]

Surface Temperature Anomalies: JJA 1816 By Member

Supplementary Figure 1: Anomalies in JJA 1816 in near surface temperature ($^{\circ}\text{C}$) and 992 hybrid level winds (m s^{-1}) for a subset of 9 of the members of the LME (runs 1-9) along with the full ensemble mean (bottom right).

Supplementary Figure 2: Anomalies in JJA 1816 in rainfall (mm day⁻¹) and 200 hybrid level winds (m s⁻¹) for a subset of 9 of members of the LME (runs 1-9) along with the full ensemble mean (bottom right).

Supplementary Figure 3: Hovmoller plot of ensemble-mean clear-sky shortwave flux through the atmosphere ($SW-C^{atm}$, left column) and clear-sky albedo (ALBDC, right column) following the 1815 eruption for land and ocean (L+O, top), land (middle), and ocean (bottom). Regions where incoming sunlight is less than 0.1 W m^{-2} or in which ocean or land are absent are shown in grey. Stippling/hatching indicate times and latitudes where the ensemble mean anomaly exceeds one/two standard deviations across the members.

Supplementary Figure 4: Zonal-ensemble-mean net TOA absorbed solar (SW, left) and outgoing longwave (FLNT, right) radiation for the 1815 eruption for all surfaces (A, B), land (C, D), and ocean (E, F), respectively. Stippling/hatching indicate times and latitudes where the ensemble mean anomaly exceeds one/two standard deviations across the members.

Supplementary Figure 5: As in **Supplementary Figure 4** but for changes in surface albedo.

Supplementary Figure 6: Evolution of ensemble zonal mean changes of ocean heat content from 0-100m by basin following the 1815 Mt. Tambora eruption, stippled where the mean change is less than twice the standard error across ensemble members.

Supplementary Figure 7: The magnitude of the meridional overturning streamfunction (Sv) in the LME in the ocean's upper 1000 m averaged from 1850-2005. Note the abscissa is scaled to reflect the area within each latitude band.

Supplementary Figure 8: Seasonal anomalies in low level (992-hybrid level) wind (m s^{-1}) and 2-m temperature from early 1815 to mid-1817.

Supplementary Figure 9: As in **Supplementary Figure 8**, except for upper level wind (200-hybrid level) and rainfall (mm day^{-1}).

Supplementary Figure 10: The forced response in cloud percent by cloud type (total/high/low/mid) for JJA 1816. Stippling indicates anomalies where the ensemble mean is greater than twice the standard error and is therefore distinguishable from internal variability at 95% confidence.

Supplementary Figure 11: Zonal-mean change in clear sky albedo following the 1815 Tambora eruption, a 2085 analogue, and their difference, stippled where the future response is distinct from that in 1816 based on twice the ensembles' standard error.

Supplementary Figure 12: As in Supplementary Figure 11 but for changes in TOA net SW flux.

Supplementary Figure 13: As in **Supplementary Figure 11** but for changes in TOA outgoing LW flux (OLR).

Supplementary Figure 14: As in Fig. **Supplementary Figure 11** but for changes in surface albedo.

Supplementary Figure 15: Nineteenth century (1800 to 1899) mean state surface wind speed (m s^{-1} , colors) and mean near-surface winds (vectors) by season (A-D) and changes by the late 21st century (2085 to 2094 minus 1800 to 1899) under RCP8.5 (E-H).

Supplementary Figure 16: Nineteenth century (1800 to 1899) mean state mixed layer depth (colors) and mean near-surface winds (vectors) by season (A-D) and changes by the late 21st century (2085 to 2094 minus 1800 to 1899) under RCP8.5 (E-H).

Supplementary Discussion

The climate response to the 1815 eruption of Mt Tambora is highly transient, evolving as a strong function of location and time. It is also accompanied by significant internal variability that limits determinations regarding regional responses and their changes in a warmer climate from the ensemble explored here. The supplementary material included illustrates these aspects, showing the broad spread across individual ensemble members in winds and temperature, and precipitation (Supplementary Figures 1, 2, respectively).

The time-varying character of radiative perturbations is an important aspect of the evolving forced climate response and is shown in Supplementary Figure 3 through Supplementary Figure 5 for SW- C^{atm} , clear sky albedo (ALBDC), TOA SW and LW fluxes, and surface albedo to provide context for both the latitudinal structure and temporal duration of the eruption's radiative effects. The radiative impacts of increases in snow cover and sea ice extent in the winters following the eruption can be inferred from anomalies in Supplementary Figure 5, to which they are the main contributor.

The supplementary figures also document changes in the ocean and relationships to its base state, with the impact on upper ocean heat content (OHC) shown in Supplementary Figure 6. The latitudinal structure of the global ocean overturning circulation is provided for context in interpreting OHC anomalies in Supplementary Figure 7.

The spatial structure of the forced response (18-member ensemble mean) in temperature, rainfall, and winds by season is shown in Supplementary Figures 8, 9 to provide context for the evolving forced response. The forced response in clouds as a function of height is shown in Supplementary Figure 10 to provide additional context for Table 1.

The radiative climate response to a simulated hypothetical 2085 eruption of Mt. Tambora, with identical aerosol emissions, is summarized and provided with context in Supplementary Figures 11 through 16. Changes in clear sky albedo (Supplementary Figure 11), shortwave (Supplementary Figure 12) and longwave (Supplementary Figure 13) top of atmosphere fluxes are largely indistinguishable from the 1815 eruption, while surface albedo (Supplementary Figure 14) reflects the poleward displacement of snow cover and sea ice in the mean state. Changes in the surface features of the response are consistent with changes in the background climate state and a more stable upper ocean that may contribute to the enhanced future response driven by weakened surface winds (Supplementary Figure 15) and shallower mixed layer depths generally (Supplementary Figure 16).